# Conformational Preferences and Antiproliferative Activity of Peptidomimetics Containing Methyl 1′-Aminoferrocene-1-carboxylate and Turn-Forming Homo- and Heterochiral Pro-Ala Motifs

**DOI:** 10.3390/ijms222413532

**Published:** 2021-12-16

**Authors:** Monika Kovačević, Mojca Čakić Semenčić, Kristina Radošević, Krešimir Molčanov, Sunčica Roca, Lucija Šimunović, Ivan Kodrin, Lidija Barišić

**Affiliations:** 1Department of Chemistry and Biochemistry, Faculty of Food Technology and Biotechnology, University of Zagreb, 10000 Zagreb, Croatia; monika.kovacevic@pbf.hr (M.K.); mcakic@pbf.hr (M.Č.S.); lsimunovic@pbf.hr (L.Š.); 2Department of Biochemical Engineering, Faculty of Food Technology and Biotechnology, University of Zagreb, 10000 Zagreb, Croatia; kradosev@pbf.hr; 3Division of Physical Chemistry, Ruđer Bošković Institute, 10000 Zagreb, Croatia; Kresimir.Molcanov@irb.hr; 4NMR Centre, Ruđer Bošković Institute, 10000 Zagreb, Croatia; sroca@irb.hr; 5Department of Organic Chemistry, Faculty of Science, University of Zagreb, 10000 Zagreb, Croatia

**Keywords:** antiproliferative activity, chirality, conformational analysis, density functional theory (DFT), ferrocene, hydrogen bonds, peptidomimetic, X-ray

## Abstract

The concept of peptidomimetics is based on structural modifications of natural peptides that aim not only to mimic their 3D shape and biological function, but also to reduce their limitations. The peptidomimetic approach is used in medicinal chemistry to develop drug-like compounds that are more active and selective than natural peptides and have fewer side effects. One of the synthetic strategies for obtaining peptidomimetics involves mimicking peptide α-helices, β-sheets or turns. Turns are usually located on the protein surface where they interact with various receptors and are therefore involved in numerous biological events. Among the various synthetic tools for turn mimetic design reported so far, our group uses an approach based on the insertion of different ferrocene templates into the peptide backbone that both induce turn formation and reduce conformational flexibility. Here, we conjugated methyl 1′-aminoferrocene-carboxylate with homo- and heterochiral Pro-Ala dipeptides to investigate the turn formation potential and antiproliferative properties of the resulting peptidomimetics **2**–**5**. Detailed spectroscopic (IR, NMR, CD), X-ray and DFT studies showed that the heterochiral conjugates **2** and **3** were more suitable for the formation of β-turns. Cell viability study, clonogenic assay and cell death analysis showed the highest biological potential of homochiral peptide **4**.

## 1. Introduction

Despite their enormous biological importance and drug-like properties, the medical use of peptides is still limited by their poor proteolytic stability, poor absorption, and low selectivity. Peptidomimetics, that is “*compounds*
*whose essential elements (pharmacophore) mimic a natural peptide or protein in 3D space and which retain the ability to interact with the biological target and produce the same biological effect*” [1] are an efficient answer to these drawbacks. In the last four decades, the concept of peptidomimetics, i.e., the art of transforming peptides into drugs, has emerged as the powerful tool in medicinal chemistry [2].

One of the synthetic approaches in the development and optimization of peptidomimetics is based on mimicking the peptide secondary structures (α-helices, β-sheets or turns) involved in protein-protein interactions (PPIs) [3]. Within a cell, PPIs form an “interactome”, an intricate network involved in physiological and pathological processes such as signal transduction, cell proliferation, growth, differentiation, apoptosis, etc. Protein-protein interfaces have bioactive “hotspots” consisting of four to eight amino acid segments, and half of them are arranged in turns [4]. The development of mimetics of PPIs “hotspot” regions that act as modulators or inhibitors of PPIs is a promising strategy for drug discovery [5].

Turns are the protein sites mainly composed of Asn, Gly and Pro where the polypeptide chain folds back on itself, making the proteins compact and globular. Since the turns are usually located at the protein surface, they are exposed to cell receptors and therefore involved in biological interactions [6]. Depending on their length and hydrogen bonding pattern, turns are classified as α-(13-membered hydrogen bonded (HB) ring), β-(10-membered HB ring) and γ-turns (7-membered HB ring). Recently, Trabocchi and Lenci [2] reviewed several conceptually different synthetic toolboxes for the design of β-turns that are involved in numerous biological recognition processes, such as peptide-antibody interactions, and recognition between peptide ligands and proteins. The turn inducing elements approach is based on the replacement of the amino acid at *i* + 1 and/or *i* + 2 with an element that both induces the formation of the turn and reduces the conformational flexibility, while the small molecular scaffolds as structural mimetics approach involves the replacement of the entire peptide backbone with the rigid scaffold that allows the alignment of the side chains in a spatial arrangement corresponding to the peptide turn residues.

Using the first approach, our group has made serious efforts in the synthesis and conformational analysis of ferrocene-containing peptidomimetics [7,8,9,10,11,12,13,14]. Due to the distance between the cyclopentadienyl (Cp) rings of 3.3 Å, the peptide chains when attached to the 1,1′-disubstituted ferrocene templates, i.e., -NH-Fn-CO- and -NH-Fn-NH- (Fn = ferrocenylene), come close enough to form 12- (**I**) [7,8,9,10,11,12] and 14-membered interstrand hydrogen-bonded rings (**II**) [13,14], respectively, in symmetrically disubstituted ferrocene peptides (Figure 1). We have therefore shown that 1,1′-disubstituted ferrocenes are capable to nucleate β-turns and β-sheet-like structures upon conjugation with amino acids and short peptides.

In 2018, Moriuchi et al. gave a review of previously synthesized symmetrically disubstituted ferrocene-dipeptide conjugates that adopt β- and γ-turn-like structures [15].

To investigate whether the asymmetrically disubstituted ferrocene conjugates with amino acids are involved in hydrogen-bonded turns, we first prepared the conjugates of methyl 1′-aminoferrocene-carboxylate with Ala (**III**) [16] and Pro (**IV**) [17] (Figure 2). Detailed spectroscopic analysis revealed two different conformational patterns consisting of seven-membered intra- (**A**, γ-turn) and nine-membered interstrand hydrogen bonded rings (**B**) in the Ala-dipeptides **III**, whereas the Pro-dipeptides **IV** adopted pattern **A** only. It was found that the different bulkiness and basicity of the Boc and Ac group affected the hydrogen-bonding patterns to some extent.

Our previous studies have shown that even the monosubstituted conjugates between the ferrocene and chiral amino acids or short peptides may induce a different sign in the circular dichroism spectra (CD) near the absorption maximum of a ferrocene chromophore (around 470 nm) [18,19,20]. We have pointed out a strong correlation between the sign of the Cotton effect and the sign of the dihedral angle between two planes, one containing the cyclopentadienyl ring and the other containing amide bond [18,19,20]. Predominance of a specific conformer arises from the amino acid sequence, which triggers a different helicity of the folded peptide through intramolecular interactions, primarily hydrogen bonds. The second substituent connected to the opposite cyclopentadienyl ring adds additional hydrogen bond donor and acceptor sites, resulting with more rigid structure through the subsequent formation of intra-or interstrand hydrogen bonds (HBs) inducing the helical chirality of the ferrocene moiety by restricting the torsional twist about the Cp(centroid)-Fe-Cp(centroid) axis. The helical arrangement (*M*- or *P*-) of the ferrocene moiety depended on the chirality of the bound amino acids or peptides. In addition, the peptide sequence, backbone homo-or heterochirality, hydrogen bond-acceptor potential of the *N*-terminal groups, and hydrogen-bonding patterns were found to regulate the size of the hydrogen-bonded ring, i.e., the type of turn.

Therefore, we decided to investigate the conformational consequences of introduction of additional hydrogen bond donor and acceptor via Ac/Boc-l-Pro and Ac/Boc-d-Pro sequence at the *N*-terminus of peptide 1. Considering that (i) the most favourable conformation of Pro is in a tight turn [21] and (ii) Pro-Xaa sequence is recognised as a β-turn sequence [22,23,24], we naturally expected an altered conformational space with more complex intramolecular hydrogen bond (IHB) patterns of the new conjugates **2**–**5** in comparison with previously synthesized compounds with only one amino acid.

The antitumor activity of ferrocenes was first reported in 1978 [25]. Since then, various ferrocene compounds have been investigated as candidates for anticancer, antibacterial, antifungal and antiparasitic drugs [26,27]. A review of the anticancer activity of ferrocene hybrids with amino acids/peptides, azoles, chalcones, coumarins, indoles, steroids, sugars, etc. was recently given by Wang et al. Due to the ability of drug-amino acid/peptide hybrids to overcome multi-drug resistance in chemotherapy and bind to specific receptors expressed on cancer cells, ferrocene-amino acid/peptide hybrids could be used to identify new anticancer agents [28]. With this in mind, we decided to investigate conjugates **2**–**5** for their antitumor activity.

## 2. Results and Discussion

### 2.1. Synthesis of Peptides ***2***–***5***

The previously established simple and efficient synthetic route to ferrocene-containing peptides [13,14,16,17] was applied here to obtain peptides **2**–**5** (Figure 1). Boc-deprotection of Boc-l-Ala-NH-Fn-COOMe **1** [16] in the presence of gaseous HCl gave a hydrochloride salt, which was processed with an excess of NEt_3_ to give the free amine required for the coupling step. Then, *C*-activated Boc-l-Pro-OH and Boc-d-Pro-OH were added to the unstable amine to obtain diastereomeric Boc-peptides **2** and **4**, respectively. Conversion of carbamates **2** and **4** to acetamides **3** and **5** was accomplished by (i) acidic Boc-deprotection and (ii) Ac-protection in the presence of acetyl chloride [14]. The characterization data with IR, NMR, and MS spectra of conjugates **2**–**5** can be found in the Appendix A.

### 2.2. Computational Study

As we have already mentioned, previously investigated conjugates **III** [16] and **IV** [17] show two different HB patterns, the one including 7-membered intrastrand and 9-membered interstrand HB rings in Ala amino acid derivatives **III**, and other with only one 7-membered ring in Pro amino acid derivatives **IV** (Figure 2). Without changing the ester substituent on one Cp ring, we can simply modify the type and the number of hydrogen bond donor and acceptor groups by incorporating an additional amino acid, e.g., Pro, to test the robustness of the existing hydrogen bond patterns. We expected formation of 10-membered rings because Pro-Xaa sequence is known as good β-turn-inducer.

A detailed conformational analysis of four newly synthesized compounds was performed hierarchically, starting from molecular mechanics and finishing with optimization of the most stable conformers in implicitly modelled solvent (SMD) with the B3LYP-D3 functional and 6 − 311 + G(d,p) basis set, LanL2DZ for iron. Additionally, all hydrogen bonds were confirmed based on the Quantum theory of atoms in molecules (QTAIM) analysis of the bond critical points between hydrogen bond acceptors and hydrogens (more details in Materials and Methods). The results of the computational study are displayed in Figure 3, Figure 4 and Figure 5 and in Appendix A.

The heterochiral d-Pro-l-Ala sequence in compounds **2** and **3** promotes formation of two hydrogen bonds (pattern **A**). The one is an intrastrand NH_Fc_⋅⋅⋅O=C_Boc/Ac_ HB, which is exactly a 10-membered ring (β-turn) as expected from Pro-Ala sequence. The other is interstrand NH_Ala_⋅⋅⋅O=C_COOMe_ HB that forms a 9-membered ring. The same IHB pattern labelled as pattern **A** is formed no matter of a relative orientation of the second Cp ring because ester group can freely rotate to accommodate best position on both sides to establish the same type of the interstrand hydrogen bond, NH_Ala_⋅⋅⋅O=C_COOMe_, in conformers with both type of helicity (*P*- in **2**–**1**, **2**–**2**, **3**–**1** and **3**–**4**; while *M*- in **2**–**3**, **3**–**2** and **3**–**3**). Many energetically close conformers differ only by puckering modes of pyrrolidine ring.

In comparison, homochiral l-Pro-l-Ala sequence favours formation of *M*-helical peptides and it is more dependent on the type of the protecting group attached to *N*-terminus. The most stable conformer of both **4** and **5** has an appropriate relative arrangement of both substituents to form interstrand 9-membered ring connected through NH_Ala_⋅⋅⋅O=C_COOMe_ hydrogen bond (IHB pattern **B**). However, bulkier *tert*-butyl (Boc-protection) group in comparison with smaller methyl (Ac-protection) prevents formation of an additional 10-membered ring, described as IHB pattern A that is observed in **4**–**1**, but not in other conformers of **4** and **5**. The other, less stable conformers of **4** and **5** are folded under the influence of the same types of NH_Ala_⋅⋅⋅O=C_COOMe_ (IHB pattern B) and NH_Fc_⋅⋅⋅O=C_Boc/Ac_ (IHB pattern C) hydrogen bonds, but only when acting individually.

### 2.3. IR Spectroscopy

Our next goal was to connect the results of the computational analysis with experimental data, especially with the determined HB patterns predicted by DFT in peptides **2**–**5**. HBs have a significant effect on the IR spectrum causing a red shift and increase in the intensity of the X–H stretching frequency when X–H∙∙∙Y hydrogen bonds are formed [29,30]. Therefore, a closer look at the amide A region in the IR spectra of the studied compounds revealed the presence of free (~3420 cm^−1^) and associated (~3300–3325 cm^−1^) NH groups, and the red-shifted stretching frequencies of the CO_Ac/Boc_ groups strongly suggest their participation in HBs [31] (Figure 6a,b and Appendix A). The ratios of free and associated NH bands in homochiral peptides **4** and **5** (~1:1.5) (Figure 6b) and heterochiral peptides **2** and **3** (~1:2.4) (Figure 6a) suggest a higher extent of hydrogen bonding in heterochiral conjugates. The intramolecular nature of the hydrogen bonds was proved by the concentration-independent IR spectra (Figure 6a,b). Otherwise, the concentration dependence of the IR spectra, i.e., the decrease in the intensities of intermolecularly engaged NH groups would be observed upon dilution. The domination of associated NH groups was also observed in the solid state of compounds **2**–**5** (Figure 6c and Appendix A).

### 2.4. NMR Spectroscopy

NMR spectroscopy is a powerful tool for studying the structure and interactions of peptides and proteins, allowing the identification of bioactive conformations responsible for their drug-like properties. An overview of NMR methods for obtaining the 3D structure of small, unlabelled peptides was recently provided by Vincenzi et al. [32]. Here, we performed detailed 1D (^1^H, ^13^C) and 2D NMR studies (^1^H-^1^H COSY, ^1^H-^1^H NOESY, ^1^H-^13^C HMQC, and ^1^H-^13^C HMBC) to assign the proton resonances and determine the individual hydrogen bonds and their strength.

Due to the hydrogen bonding deshielding, the resonances of the involved amide protons are downfield shifted (δ > 7 ppm), and the higher values of the chemical shifts indicate stronger hydrogen bonding [33]. Since the NH_Fn_ resonances of the tested peptides are significantly downfield shifted (δ ~ 8–8.6 ppm) compared to NH_Ala_ (δ ~ 6.9–7.2 ppm), their participation in stronger HBs is suggested (Figure 7, Appendix A).

Next, concentration-, temperature- and solvent-dependent NMR spectroscopy was performed to obtain further details about the conformational space of the tested conjugates. NH_Fn_ and NH_Ala_ showed no significant upfield shifts (*δ* < 0.1 ppm) at high (50 mM) vs. low concentrations (1 mM) (Figure 7), supporting their involvement in IHBs, as suggested by the concentration-independent IR data (Figure 6a,b).

We have further estimated the stability of the intramolecularly hydrogen-bonded structures in peptides **2**–**5** by examining the temperature dependence of the shifts of the amide protons. A smaller shift corresponds to a more stable conformation [34]. The signals of NH_Fn_ and NH_Ala_ of heterochiral peptide **3** showed the smallest upfield shifts compared to those of peptides **2**, **4**, and **5** (with the exception of NH_Ala_ from **4** which was slightly downfield shifted) (Figure 8). Therefore, the heterochiral Ac-peptide **3** is expected to adopt the most stable conformations compared to its counterparts, where both NHs are involved in relatively strong hydrogen bonds.

Temperature coefficients (Δδ/Δ*T*), i.e., the variations in the chemical shifts of the amide protons with temperature, imply if the NH groups are exposed to or shielded (hydrogen bonded) from the solvent, and therefore provide information about hydrogen bonding. While low Δδ/Δ*T* values (−2.4 ± 0.5 ppb K^−1^) correspond to either shielded protons which were initially downfield shifted, or protons exposed to CDCl_3_, larger Δδ/Δ*T* values always reflect initially shielded NH protons exposed during unfolding of intramolecularly hydrogen-bonded structures or dissociation of aggregates upon heating. [13,14,35,36,37,38,39,40,41,42,43] Since the concentration-independent IR and NMR spectra excluded the self-assembly, the observed larger temperature coefficients are an additional confirmation of the intramolecularly folded conformations in peptides **2**–**5**.

The multiple resonances of the amide protons at lower temperatures indicate a slow *cis*/*trans* isomerization of the proline imide bond [37,44,45]. At higher temperatures, the rate of isomerization increases, and the signals of the amide protons involved in weak hydrogen bonds coalesce. Conversely, the slow proline isomerization and decreased coalescence occur at high temperatures when the amide protons are involved in strong HBs that can induce isomer locking [46].

The multiple resonances observed for NH_Fn_ and NH_Ala_ of peptides **2**, **4** and **5** at 258 K and the coalescence that occurred upon subsequent heating to 328 K are consistent with their involvement in weak HBs. Also, the absence of multiple resonances of the amide protons of the heterochiral Ac-peptide **3** at lower temperatures is an additional confirmation of its involvement in strong IHBs and hydrogen-bond-induced folding into a stable turn structure (see Appendix A).

The chemical shifts of the four NH_Fn_ and four NH_Ala_ showed different variations upon titration of CDCl_3_ solution with DMSO-*d_6_* (Figure 9). Although these protons are involved in IHBs, they showed downfield shift with increasing DMSO content, which could be due to exposure to the hydrogen-bond-accepting solvent. The NH_Ala_ of the tested peptides showed a high degree of solvent sensitivity (Δδ ~ 0.94–1.46 ppm), which is probably due to their involvement in weak HBs. However, increasing the DMSO content from 0–56% affected the NH_Fn_ of heterochiral peptides **2** and **3** much less (Δδ ~ 0.3–0.45 ppm) than those of homochiral peptides **4** and **5** (Δδ ~ 1 ppm), indicating their involvement in stronger IHBs and folding into more stable turn structures.

As for the *cis*/*trans* proline isomerization, it is expected that when a nonpolar or less polar solvent is used, the *trans* isomer is more pronounced [14,17,47,48]. Moreover, the *trans* isomer will dominate when the IHBs are formed [14,17,49]. When CDCl_3_ solutions of Boc-peptides **2** and **4** were titrated with DMSO, the population of the *cis* isomer almost reached the amount of the *trans* fraction, which could be indicative of their participation in weaker HBs. However, the addition of DMSO had a much smaller effect on heterochiral Ac-peptide **3**, which retained almost exclusively *trans* fraction in the presence of DMSO due to its involvement in strong IHBs (Appendix A).

Two-dimensional NOESY spectroscopy was performed to further investigate the folded conformations of peptides **2**–**5** (Figure 10). We were focused on intrastrand NOE interactions between NH_Fn_ and NH_Ala_ with the *N*-terminal Boc or Ac group and on interstrand NOE interactions between NH_Fn_ and NH_Ala_ and the ester methyl group.

The observed NOE contacts between NH_Fn_ and Ac or Boc methyl protons of heterochiral peptides **2** and **3** clearly indicate the presence of intrastrand hydrogen bonds NH_Fn_···O=C_Boc/Ac_ corresponding to β-turns. NOE contact between NH_Ala_ and the ester methyl group of peptides **3** is observed, indicating the presence of interstrand hydrogen bond NH_Ala_···O=C_COOMe_ as in IHB pattern A with two hydrogen bonds (Figure 5).

As can be seen from the above 1D NMR data, the homochirality of the peptide backbone affects the conformational behaviour and makes the hydrogen bonding more sensitive and weaker compared to heterochiral analogues. The absence of NOE contact between NH_Fn_ and Boc methyl protons for homochiral peptide **4** suggests the presence of weaker HBs and lower degree of chiral organization compared to heterochiral peptides **2** and **3**. The NOEs between NH_Fn_/NH_Ala_ and Ac-methyl protons of homochiral peptide **5** were not observed, indicating that the intramolecular hydrogen bonding in **5** is very weak and its structure is not helically ordered. These observations confirm the results of the computational study. Heterochiral peptides **4** and **5** equilibrate mostly between the conformations having either IHB pattern **B** or **C**, both with one hydrogen bond, that makes them more flexible for different orientation of the Boc/Ac protecting groups relative to NH. Based on NMR study, hydrogen bonding pattern **A** with two simultaneous intra- and interstrand HBs (Figure 5) is found only in heterochiral conjugate **3** which is therefore expected to have the most ordered chiral surrounding compared to its homologues.

### 2.5. CD Spectroscopy

Recent reviews describe CD (circular dichroism) as one of the most useful techniques for measuring conformational changes in the secondary and tertiary structures of peptides and proteins during aggregation, thermal or chemical unfolding, and ligand binding interactions [50,51].

When the ferrocene scaffold is inserted into a chiral peptide chain, the formation of turns stabilized by hydrogen bonds occurs, and β-sheet-like structure is formed. Consequently, restriction of the free rotation of the ferrocene rings gives rise to helical chirality of the ferrocene core and Cotton effects in the region of ferrocene-based transitions around 480 nm. The positive Cotton effects correspond to the *P*-helicity of the ferrocene unit, while *M*-conformers induce negative Cotton effects. The most pronounced CD activity (M*_θ_* ~ 700,000 deg cm^2^ dmol^−1^) was measured for symmetrically disubstituted β-sheet-like mimetics **II** [13,14] composed of homo- and heterochiral Ala-Pro dipeptides bound to an -NH-Fn-NH- template and was attributed to a highly ordered chiral environment. Their conformational stability, realized by two strong interstrand hydrogen bonds, was confirmed by the preservation of more than 70% of CD activity upon titration with DMSO [14]. However, the noticeable loss of CD activity for asymmetrically disubstituted ferrocene peptides **III** [16] and **IV** [17] (M*_θ_* ~ 500–800 deg cm^2^ dmol^−1^) is attributed to the reduction and weakening of hydrogen bonds. We have shown that even monosubstituted aminoferrocene incorporated at the *C*-terminus of di- and tripeptide sequences can sense the chiral environment (*M**_θ_* ~ 1000–2000 deg cm^2^ dmol^−1^) resulting from the turn structures established in the attached peptide fragment [19,20].

The CD silent ferrocene region in the spectrum of homochiral peptide **5** (Figure 11) confirms the absence of chiral order predicted by NMR. Since conformational analysis indicated the presence of the folded structures in peptides **2**–**4,** they were expected to show CD activity, but with different sign depending on their homo-or heterochirality. The negative Cotton effect (M*_θ_* ~ 2500 deg cm^2^ dmol^−1^) for homochiral peptide **4** indicated *M*-helicity, whereas heterochiral peptides **2** and **3** showed almost 2-fold stronger positive Cotton effects (M*_θ_* ~ 4200–4800 deg cm^2^ dmol^−1^) related to *P*-helicity of the ferrocene core (Figure 11a). These findings are in the agreement with the computational study. The most stable conformers of **2** and **3** have a significant contribution of the *P*-1,2′ helical conformations (Appendix A) in the total population, thus resulting with a positive Cotton effect. However, *M*-1,1′ helical structures were determined to be the most abundant in derivatives **4** and **5**, especially in derivative **5** where all of the most stable conformers adopt this type of helicity. Smaller values of the pseudotorsion angles (due to 1,1′ relative orientation of the substituents on the opposite Cp rings) will result with a smaller intensity in CD spectrum. However, the conformers *M*-1,2′ that coexist in **4** will additionally enhance the negative Cotton effect.

As shown in our previous studies [14,18,19,20], the increase and strengthening of hydrogen bonding in peptides **2** and **3** leads to a higher degree of chiral organization compared to their counterpart **4** and analogues **III** [16] and **IV** [17]. It was expected that heterochiral peptides **2** and **3**, which showed less DMSO-induced shifts of NH_Fn_ compared to homochiral peptides **4** and **5**, would adopt a more stable turn conformations realized by intrastrand hydrogen bonds. However, it was found that all the tested peptides lost their CD activity in the presence of 20% DMSO, confirming our previous finding that the less stable conformations were generally formed by hydrogen bonding within the same strand [16,17,18]. Considering the hydrogen bonding potential of water (both as a donor and acceptor of hydrogen bonds), the changes in the hydrogen-bonding pattern of the peptides studied here are also expected in the water environment.

The CD activity of the peptides **2**–**5** in the solid state resembles their behaviour in the solution in terms of the sign of the Cotton effects. Here, the intensity of CD activity of heterochiral peptide **3** is much more pronounced compared to those of peptides **2**, **4,** and **5** than in the solution state, indicating the presence of a more stable turn conformation in the solid state (Figure 11b).

### 2.6. X-ray Crystal Structure Analysis

We applied the same crystallization procedure for all goal compounds, i.e., recrystallization from a solution of dichloromethane, chloroform, and ethyl acetate, but only compounds **2** and **5** gave single crystals of suitable quality for X-ray structural analysis.

The conformation in the solid state is usually affected by the anisotropic environment (i.e., the crystal field) and specific intermolecular interactions forming in the crystals, and often differs from the conformation in solution. The formation of medium-strong N–H∙∙∙O hydrogen bonds (energies in the range 5–10 kcal mol^−1^) is particularly favorable, and their energies (in the range of 5–10 kcal mol^−1^) are high enough to affect the molecular conformation.

Compound **2** adopts the bent conformation with intermolecular hydrogen bond N1–H1∙∙∙O3 (Figure 12), which is consistent with pattern A shown in Figure 5. It is probably supported by a weaker C5–H5∙∙∙O3 hydrogen bond (Appendix A). The other proton donor, N2, is oriented outward and participates in intermolecular hydrogen bonding with the carbonyl oxygen O1 of a neighbouring molecule related by translation (Figure 12, Appendix A); thus, chains parallel to the crystallographic direction [100] (i.e., parallel to axis a, Appendix A) are formed.

The conformation of compound **5** is also bent, but without intramolecular N–H∙∙∙O hydrogen bonding (Figure 12). It crystallises as a monohydrate, so the water molecule O6 interferes with the hydrogen bonding: it acts as a proton acceptor for the group N1-H1 (Appendix A). This gives rise to zig-zag chains extending in the direction [10] (i.e., crystallographic axis *b*, Appendix A). The bent conformation of **5** is stabilized by a single weak hydrogen bond C19–H19∙∙∙O2 (Figure 12, Appendix A).

The reported conformations differ from those determined by the computational study, although β-turn prevails in heterochiral derivative **2**. However, the energy penalty connected with a reorganization of the individual molecule from the most stable conformer (as determined by computational study) to less stable conformation (as determined by X-ray analysis) is usually overcome by favourable intermolecular interactions that additionally stabilize molecules in crystal [16].

### 2.7. Biological Evaluation

A further step after the successful synthesis and characterization of the investigated peptidomimetics is the determination of their biological activity. Based on the literature data and our previously published work [13,17], peptidomimetics **2**–**5** are expected to possess antiproliferative and/or antitumor activity. Therefore, the biological activity of **2**–**5** was evaluated based on their ability to inhibit the growth of MCF-7 and HeLa carcinoma cells. The cytotoxicity of the synthesized compounds was measured using the CellTiter 96^®^ AQ_ueous_ One Solution Cell Proliferation Assay. Results are expressed as cell viability (%) of treated cells versus control, non-treated cells and shown in Figure 13a,b.

All tested compounds **2**–**5** have inhibitory effects on HeLa and MCF-7 cell lines at concentrations of 100 μM and higher, as shown in Figure 13a,b, respectively. The effect of the tested peptidomimetics on cell viability is dose-dependent, i.e., the growth inhibition is proportional to the increase in the concentration of the tested compound. Boc-protected peptides **2** and **4** have stronger inhibitory effect compared to Ac-peptides **3** and **5**. Viability of cells treated with the highest concentration (500 μM) was from 35.4377% (HeLa) to 54.3296% (MCF-7) for **2** and 37.2897% (HeLa) to 48.3693% (MCF-7) for **4**. In vitro cytototoxicity results are quantified as the IC_50_ value (the half-maximal inhibitory concentration), which is defined as the concentration of the test compound that results in 50% inhibition of cell growth. These values for both cell lines and the four compounds tested were calculated from the best-fitted equations of dose-response curves and are shown in Table 1.

According to the IC_50_ values, compound **4** has the strongest inhibitory effect on MCF-7 cells, with slightly less pronounced impact on HeLa cells. For compound **2**, an IC_50_ value was calculated only for HeLa cells, while no IC_50_ values were calculated for other compounds from the experimental data. In the range of tested concentrations (10 μM−500 μM), no 50% inhibition of cell growth was observed, so for compounds **2**, **3** and **5** on MCF-7 cells, and for compounds **3** and **5** on HeLa cells, the IC_50_ value can be considered higher than 500 μM. The cytotoxicity assay revealed that HeLa cells are somewhat more sensitive to the effect of the tested peptidomimetics **2**–**5**, therefore the remaining experiments were performed on HeLa cells.

Another way to assess cell survival and determine the efficacy of cytotoxic agents, is a colony formation test or clonogenic assay, *in vitro* method based on the ability of a single cell to grow into a colony. After treatment with the tested compounds, the surviving cells take about 1–3 weeks to form colonies, but only a small fraction of exposed cells retain the ability to form colonies [52]. Since clone formation is in some respects a property of unlimited growth, which is a special feature of tumor cells, the clonogenic assay may serve as a good indicator of the antitumor potential of the test compounds. Therefore, peptides **2**–**5** were also analyzed by a clonogenic assay on HeLa cells treated with 100 μM and 500 μM concentrations of **2**–**5**. After 17 days of in vitro cultivation, colonies became visible and were then coloured with 0.5 % crystal-violet, counted, and photographed (see Appendix A).

Based on the number of colonies counted, the plating efficiency (PE) and surviving fraction (SF) were calculated for compounds **2**–**5** (Table 2) according to the equations in the protocol of Franken et al. [53]. A higher SF value means that a higher colony forming ability is maintained after treatment with the test substance, which could be related to less pronounced cytotoxic efficacy of that compound.

When HeLa cells were treated with 100 μM of the tested compounds, higher values of PE were calculated for Ac-peptides **3** and **5** than for Boc-peptides **2** and **4**, while survival of cells treated with 500 μM was seen only after treatment with compound **5**. The results of the clonogenic analysis are consistent with their cytotoxicity, as there is a significant difference in the number of visible colonies grown after treatment with 100 μM of Boc-peptides **2** and **4**, in contrast to Ac-peptides **3** and **5**, which showed a weaker effect on the growth and survival of HeLa cells.

The observed cytotoxicity of the tested peptidomimetics could be the result of their impact on two basic cell processes, cell division and/or cell death. Programmed cell death through the process of apoptosis was originally defined based on morphological characteristics, so the initial identification of apoptotic cells is often observed under the microscope. Control and treated HeLa cells were stained with the fluorescent dyes fluorescein diacetate (FDA) and propidium iodide (PI), examined, and photographed under the EVOS FLoid Cell Imaging Station fluorescence microscope, as shown in Figure 14.

The use of FDA and PI fluorescent stain allows differentiation between living, necrotic, and apoptotic cells in the sample of treated cells. In a photograph of control HeLa cells, there are plenty of live, green-stained cells, whereas in the case of HeLa cells treated with compounds **2** and **4**, respectively, only a few dead cells stained red can be seen (Figure 14). To further confirm and quantify the observed morphological changes indicative of the induction of cell death during treatment with the tested peptides **2**–**5,** flow cytometry analysis was conducted using the Muse^®^ Cell Analyser and the Muse™ Annexin V & Dead Cell Kit. The results of this analysis for HeLa cells are shown in Figure 15.

Double staining with 7-aminoactinomycin D (7-AAD) and Annexin-V allows differentiation between populations of live, dead, and early/late apoptotic cells in the sample by flow cytometry. The results of the analysis show that HeLa cells treated with compound **4** (500 µM) have the highest percentage of the total apoptotic cells (33.19% ± 19,123), followed by compound **2** with a total apoptosis of 30.32% ± 3,8201 (Figure 15). The lowest percentage of total apoptotic cells was determined in a sample treated with 500 µM of compound **5** (11.45%), which had the weakest effect on HeLa cells according to all other methods used for the biological evaluation of peptides **2**–**5**. The induction of apoptosis by the action of peptidomimetics has been reported previously [53], whereby peptidomimetics with anticancer properties bind to target proteins and mimic interactions that activate specific death pathways in cancer cells, which then die by apoptotic cell death. A recent review on peptidomimetics highlights their crucial role as inhibitors of protein-protein interactions in cancer cells. Based on the traditional classification of peptidomimetics into types I to III, which are divided into distinct classes A–D [54], peptidomimetics **2**–**5** belong to class A, like type I mimetics, which are involved in apoptosis regulation through inhibitory effects on the p53–MDM2 and p53–MDMX complexes, among other modes of action [54].

Our previous work on ferrocene peptidomimetics has shown that conformational patterns do not have a decisive influence on biological activity, and that lipophilicity contributes to biological activity, i.e., ferrocene peptidomimetics with larger retention factors *(R*_f_) had better antiproliferative capacity [13,17]. The results obtained here on the increased antiproliferative activity of Boc-peptides **4** (*R*_f_ = 0.55) and **2** (*R*_f_ = 0.52) compared to the more polar Ac-peptides **3** (*R*_f_ = 0.33) and **5** (*R*_f_ = 0.35) are consistent with the above findings. Moreover, structural modification of the inactive Boc-l-Pro-NH-Fn-COOMe (**IV**) [17] by insertion of l-Ala between ferrocene core and Pro resulted with the promising outcome on Boc-peptide **4**, which could serve as a leading compound for further research and development. Considering the results obtained so far, we are encouraged to continue this study with the aim of developing an antitumor drug.

## 3. Materials and Methods

### 3.1. General Procedure and Methods

The synthesis of peptides **2**–**5** was carried out under argon atmosphere and the chemicals used for the reactions were analytically pure. CH_2_Cl_2_ used for synthesis, CD measurements and FTIR was dried (P_2_O_5_), distilled over CaH_2_, and stored over molecular sieves (4 Å). EDC (Acros Organics, Geel, Belgium), HOBt (Aldrich, Santa Clara, CA, USA) and acetyl chloride (Aldrich), were used as received. The synthesis of Boc-l-Ala-NH-Fn-COOMe (**1**) has been described previously [16]. Its *N*-terminus was deprotected by exposure to gaseous HCl. The *N*-termini of l- and d-proline were protected in the presence of sodium hydroxide, aqueous dioxane and di-*tert*-butyldicarbonate to give Boc-l-Pro-OH and Boc-d-Pro-OH, respectively. Boc-l-Pro-OH and Boc-d-Pro-OH were activated with the coupling reagent HOBt for 1 h in CH_2_Cl_2_. The products were purified by preparative thin layer chromatography on silica gel (Kieselgel 60 HF254, KGaA, Darmstadt, Germany) using EtOAc/CH_2_Cl_2_ mixture or pure EtOAc as eluent. Infrared spectra were recorded as CH_2_Cl_2_ solutions between NaCl windows or in KBr using a MB 100 mid-FTIR spectrometer (Bomem, Saint-Jean-Baptiste, Canada ) ((s) = strong, (m) = medium, (w) = weak, (br) = broad, (sh) = shoulder). The ^1^H- and ^13^C-NMR spectra were recorded at 600 MHz using an Avance spectrometer (Bruker, Rheinstetten, Germany) with a 5 mm TBI probe at the Ruđer Bošković Institute and were referenced to the peak of the residual solvent (CDCl_3_-*d*, ^1^H: δ = 7.24 ppm, ^13^C: δ = 77.23 ppm). In the case of the CDCl_3_-*d*/DMSO-*d*_6_ mixture, calibration was performed using Me_4_Si as an internal standard (^1^H: δ = 0.0 ppm). Double resonance experiments (COSY, NOESY, HMQC and HMBC) were performed to facilitate the assignment of signals; ((s) = singlet, (d) = doublet, (m) = multiplet, (dd) = doublet of doublets, (td) = triplet of doublets, (dq) = doublet of quartets). Unless otherwise stated, all spectra were recorded at 298 K. NMR titrations were performed by adding 10 μL portions of DMSO-*d*_6_ to NMR tubes containing CDCl_3_-*d* solutions of the peptides under study (*c* = 2.5 × 10^−2^ M). Spectra were recorded after each addition, and DMSO-*d*_6_ was added until no change in the chemical shift of the amide protons was observed. CD spectra were recorded using a model 810 spectropolarimeter (Jasco, Tokyo, Japan) in CH_2_Cl_2_ or KBr. Molar ellipticity coefficients [*θ*] are given in degrees, concentration *c* in molL^−1^, and path length *l* in cm, so that the unit for [*θ*] is deg cm^2^ dmol^−1^. Mass spectra were recorded using HPLC-MS system coupled to a triple-quadrupole mass spectrometer, operating in a negative ESI mode (Agilent, Palo Alto, CA, USA). High-resolution mass spectra were recorded using a MALDI-TOF/TOF 4800+ analyser (SCIEX Headquarters, Framingham, MA, USA). Melting points were determined using Reichert Thermovar apparatus (Reichert, Vienna, Austria).. Single crystal measurements were performed with an Xcalibur Nova R system (Oxford Diffraction, Wroclaw, Poland).

#### 3.1.1. Synthesis of Boc-d-Pro-l-Ala-NH-Fn-COOMe (**2**) and Boc-l-Pro-l-Ala-NH-Fn-COOMe (**4**)

The HCl_gas_ was purged through the suspension of Boc-l-Ala-NH-Fn-COOMe (**1**) (1000 mg, 2.32 mmol) in dry CH_2_Cl_2_ (5 mL) at 0 °C. After 30 min, the solvent was evaporated in vacuo, leaving a dark yellow hydrochloride salt, which was then suspended in CH_2_Cl_2_ and treated with NEt_3_ (pH ~ 8) to afford an unstable free amine suitable for coupling to Boc-l-Pro-OH or Boc-d-Pro-OH (998 mg, 4.64 mmol) using the standard EDC/HOBt method; EDC (1007 mg, 5.57 mmol), HOBt (753 mg, 5.57 mmol). The reaction mixtures were then stirred at room temperature until the ferrocene amine was completely consumed, which was monitored by TLC (~1 h). Standard work-up (washing with a saturated aqueous solution of NaHCO_3_, a 10% aqueous solution of citric acid and brine, drying over Na_2_SO_4_ and evaporation in vacuo) including TLC purification of the crude products (EtOAc: CH_2_Cl_2_ = 1: 5; *R*_f_ = 0.52 (**2**), *R*_f_ = 0.55 (**4**)) gave orange solids of **2** (1107 mg, 89%) and **4** (1213 mg, 92%).

Boc-d-Pro-l-Ala-NH-Fn-COOMe (**2**): m.p. = 119.2 °C. IR (CH_2_Cl_2_) ῠ_max_/cm^−1^: 3418 w (NH_free_), 3325 m (NH_assoc._), 1705 s (C = O_COOMe_), 1684 s, 1671 s (C = O_CONH_), 1557 s, 1531 s (amide II). IR (KBr) ῠ_max_/cm^−1^: 3509 w(NH_free_), 3308 m (NH_assoc._), 1714 s (C = O_COOMe_), 1695 s, 1671 s (C = O_CONH_). ^1^H-NMR (600 MHz, CDCl_3_) δ/ppm: 8.37 (s, 0.89H, NH_Fn *trans*_), 7.70 (s, 0.11H, NH_Fn *cis*_), 7.24 (d, *J* = 6.3 Hz, 0.89H, NH_Ala *trans*_), 6.66 (d, 0.11H, NH_Ala *cis*_), 5.09 (s, 1H, H-3), 4.81 (s, 2H, H-8), 4.72–4.69 (m, 2H, H-9, CH_Ala_) 4.58 (s, 1H, H-4), 4.40 (s, 1H, H-7), 4.35 (s, 1H, H-10), 4.16 (s, 1H, CH-α (Pro)), 4.01 (s, 1H, H-2), 3.95 (s, 1H, H-5), 3.79 (s, 3H, COOMe), 3.53 (td, *J* = 10.3 Hz, 6,7 Hz, 1H, CH_2_-δ (Pro)), 3.46 (s, 1H, CH_2_-δ′ (Pro)), 2.22–2.07 (m, 2H, CH_2_-β, CH_2_-β′ (Pro)), 1.89–1.86 (m, 2H, CH_2_-γ, CH_2_-γ′ (Pro)), 1.50 (s, 9H, (CH_3_)_3 Boc_), 1.48 (d, *J* = 7.30 Hz, 3H, CH_3 Ala_). ^13^C-NMR (150 MHz, CDCl_3_) δ/ppm: 172.47 (CO_Fn_), 172.10 (CO_COOMe_), 170.89 (CO_Ala_), 155.26 (CO_Boc_), 96.31 (C-1, Fn), 80.37 (C_q Boc_), 72.79 (C-7), 72.45 (C-10), 71.88 (C-6), 71.75 (C-8), 71.43 (C-9), 66.59 (C-2), 65.40 (C-5), 63.94 (C-4), 62.54 (C-3), 61.10 (C-α, Pro), 51.87 (CH_3 COOMe_), 48.59 (CH_Ala_), 47.53 (CH_2_-δ, Pro), 29.87 (CH_2_-β, Pro), 28.58 ((CH_3_)_3 Boc_), 24.96 (CH_2_-γ, Pro), 17.56 (CH_3 Ala_). ESI-MS (H_2_O:MeOH = 50:50): *m*/*z* 526.1 ((M − H)^−^). MALDI-HRMS *m*/*z* = 527.1708 (calculated for C_25_H_33_N_3_O_6_Fe = 527.1718).

Boc-d-Pro-l-Ala-NH-Fn-COOMe (**4**): m.p. = 66.9 °C. IR (CH_2_Cl_2_) ῠ_max_/cm^−1^: 3418 w (NH_free_), 3310 m (NH_assoc._), 1705 s (C = O_COOMe_), 1674 s (C = O_CONH_), 1555 s, 1503 s (amide II). IR (KBr) ῠ_max_/cm^−1^: 3505 w (NH_free_), 3296 m (NH_assoc._), 1715 s (C = O_COOMe_), 1669 s, 1660 s (C = O_CONH_). ^1^H-NMR (600 MHz, CDCl_3_) δ/ppm: 8.15 (s, 0.9H, NH_Fn *trans*_), 7.74 (s, 0.1H, NH_Fn *cis*_), 6.85 (d, *J* = 6.3 Hz, 0.9H, NH_Ala *trans*_), 6.67 (d, 0.1H, NH_Ala *trans*_), 4.81 (s, 1H, H-3), 4.76 (s, 2H, H-8, H-9), 4.60 (s, 1H, H-4), 4.48 (m, 1H, CH_Ala_), 4.38 (s, 1H, H-7), 4.37 (s, 1H, H-10), 4.34 (s, 1H, CH-α (Pro)), 4.00 (s, 2H, H-2, H-5), 3.78 (s, 3H, COOMe), 3.50–3.45 (m, 2H, CH_2_-δ, CH_2_-δ′ (Pro)), 2.22–2.16 (m, 2H, CH_2_-β, CH_2_-β′ (Pro)), 1.92–1.91 (m, 2H, CH_2_-γ, CH_2_-γ′ (Pro)), 1.49 (s, 9H, (CH_3_)_3 Boc_), 1.42 (d, *J* = 6.98 Hz, 3H, CH_3 Ala_). ^13^C-NMR (150 MHz, CDCl_3_) δ/ppm: 172,35 (CO_Fn_), 171.95 (CO_COOMe_), 170.45 (CO_Ala_), 156.34 (CO_Boc_), 95.79 (C-1, Fn), 81.24 (C_q Boc_), 73.08 (C-7), 72.97 (C-10), 71.90 (C-6), 71.41 (C-8), 71.08 (C-9), 66.56 (C-2), 66.39 (C-5), 63.15 (C-4), 62.71 (C-3), 60.97 (C-α, Pro), 51.70 (CH_3 COOMe_), 49.74 (CH_Ala_), 47.57 (CH_2_-δ, Pro), 29.14 (CH_2_-β, Pro), 28.54 ((CH_3_)_3 Boc_), 24.84 (CH_2_-γ, Pro), 17.54 (CH_3 Ala_). ESI-MS (H_2_O:MeOH = 50:50): *m*/*z* 526.1 ((M − H)^−^). MALDI-HRMS *m*/*z* = 527.1729 (calculated for C_25_H_33_N_3_O_6_Fe = 5.271.718).

#### 3.1.2. Synthesis of Ac-d-Pro-l-Ala-NH-Fn-COOMe (**3**) and Ac-l-Pro-l-Ala-NH-Fn-COOMe (**5**)

The transformation of carbamates **2** and **4** (1000 mg, 1.89 mmol) to acetamides **3** and **5** began with the acidic Boc-deprotection described above. Their free amines, obtained by treating the hydrochloride salt with NEt_3_ (2.07 mL, 23.7 mmol), were cooled to 0 °C and acetyl chloride (807 μL, 11.34 mmol) was added dropwise, stirring in an ice bath. After TLC monitoring showed complete conversion of the starting materials, the reaction mixtures were poured into water and extracted with CH_2_Cl_2_. The combined organic phases were washed with a brine, dried over Na_2_SO_4_ and evaporated to dryness in vacuo. The resulting crude products were purified by TLC on silica gel (EtOAc; *R*_f_ = 0.33 (**3**), *R*_f_ = 0.35 (**5**)) to give orange solids of **3** (1132 mg, 60%) and **5** (1213 mg, 64%).

Ac-d-Pro-l-Ala-NH-Fn-COOMe (**3**): m.p. = 132.3 °C. IR (CH_2_Cl_2_) ῠ_max_/cm^−1^: 3424 w (NH_free_), 3303 m (NH_assoc._), 1706 s (C = O_COOMe_), 1688 s, 1630 s (C = O_CONH_), 1557 s, 1541 s, 1521 m (amide II). IR (KBr) ῠ_max_/cm^−1^: 3542 w (NH_free_), 3308 s, 3259 s, 3222 m (NH_assoc._), 1714 s (C = O_COOMe_), 1690 s, 1679 s, 1688 s (C = O_CONH_). ^1^H-NMR (600 MHz, CDCl_3_) δ/ppm: 8.58 (s, 1H, NH_Fn_), 7.08 (d, *J* = 9.1 Hz, 1H, NH_Ala_), 5.05 (s, 1H, H-3), 4.78 (s, 1H, H-8), 4.76 (s, 1H, H-9), 4.70 (s, 1H, H-4), 4.62 (dq, *J* = 8.5 Hz, 7.1 Hz, 1H, CH_Ala_), 4.40 (s, 1H, H-7), 4.35 (s, 1H, H-10), 4.26 (dd, *J* = 7.7 Hz, 5,6 Hz, 1H, CH-α (Pro)), 3.98 (s, 1H, H-2), 3.95 (s, 1H, H-5), 3.77 (s, 3H, COOMe), 3.68 (td, *J* = 9.8 Hz, 6,9 Hz, 1H, CH_2_-δ (Pro)), 3.55 (td, *J* = 9.8 Hz, 6,5 Hz, 1H, CH_2_-δ′ (Pro)), 2.28–2.23 (m, 1H, CH_2_-γ Pro)), 2.21–2.18 (m, 1H, CH_2_-β (Pro)), 2.16 (s, 3H, CH_3 Ac_), 2.15–2.11 (m, 1H, CH_2_-β′ (Pro)), 2.00–1.95 (s, 1H, CH_2_-γ′ (Pro)), 1.49 (d, *J* = 7.2 Hz, 3H, CH_3 Ala_). ^13^C-NMR (150 MHz, CDCl_3_) δ/ppm: 172.08 (CO_Ac_), 172.03 (CO_COOMe_), 170.63 (CO_Ala_), 170.31 (CO_Fn_), 96.56 (C-1, Fn), 72.80 (C-7), 72.64 (C-10), 71.91 (C-6), 71.46 (C-8), 71.20 (C-9), 66.45 (C-2), 65.74 (C-5), 63.25 (C-4), 62.69 (C-3), 61.19 (C-α, Pro), 51.78 (CH_3 COOMe_), 49.11 (CH_Ala_), 48.66 (CH_2_-δ, Pro), 29.23 (CH_2_-β, Pro), 25.52 (CH_2_-γ, Pro), 22.87 (CH_3 Ac_), 17.55 (CH_3 Ala_). ESI-MS (H_2_O:MeOH = 50:50): *m*/*z* 468.1 ((M − H)^−^). MALDI-HRMS *m*/*z* = 469.1280 (calculated for C_22_H_27_N_3_O_5_Fe = 4.691.300).

Ac-l-Pro-l-Ala-NH-Fn-COOMe (**5**): m.p. = 125.1 °C. IR (CH_2_Cl_2_) ῠ_max_/cm^−1^: 3420 w (NH_free_), 3309 m (NH_assoc._), 1705 s (C = O_COOMe_), 1696 s, 1680 s, 1636 s (C = O_CONH_), 1555 s, 1540 s, 1507 m (amide II). IR (KBr) ῠ_max_/cm^−1^: 3499 w (NH_free_), 3288 m (NH_assoc._), 1714 s (C = O_COOMe_), 1673 s, 1630 s (C = O_CONH_). ^1^H-NMR (600 MHz, CDCl_3_) δ/ppm: 7.99 (s, 0.95H, NH_Fn_ *_trans_*), 7.65 (s, 0.05H, NH_Fn *cis*_), 7.22 (d, *J* = 7.1 Hz, 0.94H, NH_Ala *trans*_), 6.86 (d, 0.06H, NH_Ala *cis*_), 4.78 (s, 1H, H-8), 4.76 (s, 1H, H-3), 4.74 (s, 1H, H-9), 4.67 (s, 1H, H-4), 4.57 (m, 1H, CH-α (Pro)), 4.48 (dq, *J* = 8.4 Hz, 7,1 Hz, 1H, CH_Ala_), 4.39 (s, 1H, H-7), 4.37 (s, 1H, H-10), 4.02 (s, 1H, H-2), 3.96 (s, 1H, H-5), 3.77 (s, 3H, COOMe), 3.67–3.64 (m, 1H, CH_2_-δ, (Pro)), 3.52–3.50 (m, 1H, CH_2_-δ′ (Pro)), 2.32 (s, 1H, CH_2_-γ, Pro)), 2.18 (s, 3H, CH_3 Ac_), 2.08–2.06 (m, 2H, CH_2_-β, CH_2_-β′ (Pro)), 1.93 (s, 1H, CH_2_-γ′ (Pro)), 1.42 (d, *J* = 7.0 Hz, 3H, CH_3.Ala_). ^13^C-NMR (150 MHz, CDCl_3_) δ/ppm: 171.82 (CO_Ac_), 171.81 (CO_COOMe_), 171.66 (CO_Ala_), 170.28 (CO_Fn_), 95.76 (C-1, Fn), 72.67 (C-7), 72.65 (C-10), 72.21 (C-6), 71.43 (C-8), 71.08 (C-9), 66.62 (C-2), 66.01 (C-5), 63.04 (C-4), 62.87 (C-3), 60.65 (C-α, Pro), 51.76 (CH_3 COOMe_), 49.58 (CH_Ala_), 48.67 (CH_2_-δ, Pro), 28.68 (CH_2_-β, Pro), 25.21 (CH_2_-γ, Pro), 22.93 (CH_3 Ac_), 17.22 (CH_3 Ala_). ESI-MS (H_2_O:MeOH = 50:50): *m*/*z* 468.1 ((M − H)^−^). MALDI-HRMS *m*/*z* = 469.1280 (calculated for C_22_H_27_N_3_O_5_Fe = 469.1300).

#### 3.1.3. Computational Details

Conformational analyses of compounds **2**–**5** were done in three stages. First, a series of low-level optimizations with molecular mechanics, OPLS2005 force field, were performed in MacroModel v10.3 [55,56,57]. The most stable conformers were selected for further optimizations at a high level of theory and run in Gaussian16 [58] with a default grid and convergence criteria B3LYP/Lanl2DZ. The last stage included optimization of the most stable conformers at the B3LYP/6-311+G(d,p) (LanL2DZ basis set on Fe) level of theory while surrounding solvent (chloroform) were described as polarizable continuum (SMD) [59]. Vibrational analysis was performed to verify each structure as a minimum on the potential energy surface and the reported energies refer to standard Gibbs free energies at 298 K. QTAIM theory were used to characterize hydrogen bonds in AIMAll package [60]. Topological parameters of the displayed bond critical points between hydrogen bond acceptors and hydrogen atoms were calculated and verified according to the Koch and Popelier criteria [61,62].

#### 3.1.4. Crystallographic Study

*X*-ray diffraction: single crystal measurements were performed on a Rigaku Oxford Diffraction Xtalab Synergy S (2) and an Oxford Diffraction Xcalibur Nova R (5), using mirror-monochromated CuKα radiation. Program package CrysAlis PRO [63] was used for data reduction and numerical absorption correction.

The structures were solved using SHELXS97 [64] and refined with SHELXL-2017 [65]. Models were refined using the full-matrix least squares refinement; all non-hydrogen atoms were refined anisotropically. Rigid-body restraints were applied to ADPs of C atoms of cyclopentadienyl ring C1→C5 in **5**. Hydrogen atoms were located in a difference Fourier map and refined either as riding entities. Hydrogen atoms of water molecule O6 in 5 could not be located from the difference map and were therefore not modelled. Molecular geometry calculations were performed by PLATON [66] and molecular graphics were prepared using ORTEP-3 [67], and Mercury [68]. Crystallographic and refinement data for the structures reported in this paper are shown in in Appendix A. The crystallographic data have been deposited in the Cambridge Structural Database as entries No. 2122149 and 2122150.

#### 3.1.5. Biological Activity

Materials: Trypsin-EDTA (0.25%), FBS (fetal bovine serum) and PBS (phosphate buffer saline) were purchased from Sigma-Aldrich while DMEM (Dulbecco’s Modified Eagle Medium) was purchased from Capricorn Scientific GmbH (Ebsdorfergrund, Germany). The CellTiter 96^®^ AQ_ueous_ One Solution Cell Proliferation Assay was purchased from Promega (Madison, WI, USA). Fluorescein diacetate (FDA) and propidium iodide (PI) were purchased from Sigma-Aldrich. The Muse Annexin V Dead Cell kit was purchased from EMD Milipore Corporation (Merck KGaA).

Cell culture and cultivation conditions: Two adherent human cell lines used in this work were obtained from the Ruđer Bošković Institute (Zagreb, Croatia). The HeLa cell line derived from the cervical adenocarcinoma (ATCC No. CCL-2) and the MCF-7 cell line derived from breast adenocarcinoma (ATCC No. HTB-22) were cultured in DMEM supplemented with 5% FBS and maintained in BioLite petri dishes for cell culture (Thermo Fisher Scientific, Waltham, MA, USA) in an incubator under a humidified atmosphere and 5% CO_2_ at 37 °C. Cells in the exponential growth phase were trypsinized, counted by the trypan blue method using an improved Neubauer hemocytometer, and used to set up individual experiments. BioLite 6-well and 96-well plates were used for individual experiments to test compounds of interest (Thermo Fisher Scientific).

Evaluation of cytotoxicity: The effect of peptides **2**–**5** on cell viability was examined using the CellTiter 96^®^ AQ_ueous_ One Solution Cell Proliferation Assay, which was performed according to the manufacturer’s instructions with minor modifications and as described [69]. In brief, HeLa and MCF-7 cells were seeded in 96-well plates at a density of 3 × 10^4^ cells per well in 100 μL of media. Stock solutions of peptides 2–5 were prepared as 10 mM solutions of compounds in ethanol, sterilized by filtration through a 0.22 µM filters and then, prior to each experiment, diluted in culture medium. After overnight incubation, HeLa and MCF-7 cells were treated with peptides 2–5 at nominal concentrations ranging from 10 μM to 500 μM. After the 72 h treatment, 10 μL of CellTiter 96^®^ AQ_ueous_ One Solution Cell Proliferation reagent was added to each well, and the cells were incubated for an additional 3 h. Subsequently, absorbance was measured at 490 nm on the microplate reader (Tecan, Mannedorf,, Switzerland). Cell viability was expressed as the percentage of treated cells versus control cells. Experiments were performed three times with five parallels for each concentration of compound tested and data were expressed as mean ± SD. The corresponding IC_50_ values were calculated from the dose-response curves using equations of best-fitted trend lines.

Clonogenic assay: The clonogenic analysis began by seeding pre-cultured HeLa cells in 6-well plates at an initial concentration of 200 cells in 2 mL of culture medium per well. The cells were incubated under optimal conditions and treated with peptides **2**–**5** at a concentration of 100 μM and 500 μM after 24 h. There was also a control cells that were not treated with peptidomimetics. Three days after the cells were treated, the growth medium containing the test compounds was removed and replaced with fresh growth medium, after which the plate with the HeLa cells was returned to the incubator for further cultivation. After treatment with the test substances, the surviving cells need about 1–3 weeks to form colonies. In this work, the colonies formed were visible 17 days after initial seeding of the cells. Staining the grown colonies with crystal-violet begins by removing the growth medium and washing the cells with 1 mL of PBS buffer. Then 2.5 mL of methanol was added to fix the cells, which was removed after 10 min. The plates are then allowed to air dry completely. A 0.5% solution of crystal-violet is then added and incubated for 10 min. In the final step, the dye is removed and the colonies in the wells are rinsed with 1 mL of PBS buffer and deionized water. The number of colonies grown was then counted and the plating efficiency (PE) and survival fraction (SF) were calculated according to the equations in the protocol of Franken et al. [52]. PE is the ratio of the number of colonies to the number of seeded cells, while SF is the number of colonies formed after treatment of the cells, expressed as PE.

Analysis of cell death by fluorescence microscopy and flow cytometry: for fluorescein diacetate and propidium iodide staining, HeLa cells were seeded in 6-well plates at a concentration approximately about 1 × 10^5^ cells mL^−1^ and treated with 500 μM of peptides **2**–**5** after 24 h. After the 72 h treatment, cells were washed with PBS, trypsinized, centrifuged, and resuspended in 0.2 mL of PBS. Cell were stained with FDA and PI according to the method described by us [70] and immediately examined with the fluorescent microscope EVOS FLoid Cell Imaging Station (Thermo Fisher Scientific).

Quantitative analysis of live, apoptotic, and dead cells treated with peptides **2**–**5** was performed with the Muse Cell Analyzer (EMD Millipore Corporation, Burlington, MA, USA) using the Muse Annexin V & Dead Cell Kit according to the manufacturer’s specifications. In brief, HeLa cells were plated into a 6-well culture at a density of 5 × 10^4^ cells mL^−1^ (2 mL per well) and treated with the 500 μM concentration of conjugates **2**–**5** for 72 h. After treatment with the test compounds, both floating and adherent cells were collected, centrifuged (600 g min^−1^), and suspended in cell culture medium to adjust the cell concentration according to the manufacturer’s protocol. Then, 100 μL aliquots of the cell suspension were added to 100 μL of Muse Annexin V & Dead Cell Reagent and incubated for 20 min in the dark at RT. Cells were then analyzed using the Muse Cell Analyzer. Each compound was tested in duplicate, and each experiment was performed twice.

The data in the graphs are expressed as mean ± standard deviation (±SD), and the error bars in the figures indicate the SD. Differences between means were analyzed using the ANOVA test, followed by post-hoc Tukey’s test. A significant difference was considered at a *p* value < 0.05.

## 4. Conclusions

New insight is provided into the effects of the constituent homo- and heterochiral Pro-Ala sequences on the conformational properties of the asymmetrically disubstituted ferrocene peptidomimetics.

The results of the DFT study agree quite well with the spectroscopic (IR, NMR) data, also confirming the theoretically predicted differences in heterochiral vs. homochiral analogs. According to the complementary experimental and computational study, the heterochiral Pro-Ala sequence initiated more complex hydrogen-bonding patterns consisting of intrastrand 10-membered (β-turn) and interstrand 9-membered rings. In comparison, the homochiral Pro-Ala sequence resulted in more flexible conformations in which mostly one of these two hydrogen bonds occur.

Of the four peptides tested, Boc-peptide **4** has the strongest inhibitory effect on tumor cells HeLa and MCF-7, the greatest potential to promote apoptotic cell death in HeLa cells and the highest ability to reduce survival of treated HeLa cells. All in all, we can conclude that peptidomimetic **4**, although its IC_50_ value is quite high compared to referent antitumour drug such as cisplatin, has the potential to serve for development of new antitumor drugs. Moreover, the obtained results are certainly a valuable guide for future research on the synthesis of new peptidomimetics that are structurally similar to compound **4** and hopefully will have improved biological properties.

## Data Availability

The crystallographic data have been deposited in the Cambridge Structural Database as entries No. 2122149 and 2122150. The DFT, spectroscopic and biological evaluation data are provided as figures and tables and are included in this paper.

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
