# Peer review of "Conformational Preferences and Antiproliferative Activity of Peptidomimetics Containing Methyl 1′-Aminoferrocene-1-carboxylate and Turn-Forming Homo- and Heterochiral Pro-Ala Motifs"

_ijms, 2021, doi:10.3390/ijms222413532_

Round 1
Reviewer 1 Report
This is a very interesting research. The authors deal with peptidomimetics based on Pro-Ala motif and methyl 1'-aminoferrocene-1-carboxylate. 4 compounds are synthesized, characterized and evaluated.
The authors examine the turn formation in these structures by the means of combined DFT calculations, IR spec, ECD spec, NMR and in solid state by X-ray analysis. Additionally, the synthesized compounds are assayed for cytotoxic activity.
The spectroscopic investigations are done in a correct way. The same are the DFT calculations.
Here I would suggest adding in conclusions a sentence or two commenting on how the DFT conformations agree with the facts deduced from spectroscopy.
Besides, having the DFT conformations, the authors might attempt reproducing the NMR spectra by DFT calculations, giving an additional hint in favour of the proposed conformations and deductions. (not obligatory, but could be added value)
Do authors expect the same H-bonding pattern in polar solvents, including in one relevant to biomolecules, that is water? DMSO-d6 titrations give suggestions towards this, but a more explicit comment would be nice for the less experienced readers.
As to the cytotoxicity, this part is also done in a proper way. I would only be rather careful with descriptions like "drug-like candidate 4". By druglikeness, I believe, medicinal chemists describe a set of properties of a given molecule that render it likely that a compound would become a drug bioavailable upon oral administration (mostly). These properties are molecular weight, lipophilicity in appropriate range, number of Hbond donors/acceptors etc. etc. These structures are by no means "drug-like" in this sense. Perhaps, the authors meant that 4 may be a lead compound. But this is also some exaggeration -> 4 has visible but still very limited cytotoxic activity, so this is very overoptimistic to call it this way.
Besides, what is the solubility of these compounds? I would imagine a Boc-peptide with addition of ferrocene fragment to be not perfectly soluble in high micromolar concentrations. Would this not affect the cytotoxicity testing? In particular if there's a 72h incubation time? would not some precipitation or coagulation occur over such time, even if the sample were initially fully dissolved?
I recommend the paper be published after claryfing those minor points.
Reviewer 2 Report
The format of manuscript is not suitable for publication. The authors should constrict all paragraph of manuscript. The authors should provide the statistic results and discussion these results in the paragraph of results and discussion in the manuscript. There manuscript have a lot of typo and grammar mistakes at present stage.
Round 2
Reviewer 2 Report
- The authors should re-organize the paragraph of 3.1.5 biological activity. The section should more clearly to facilitate understanding.
- The authors should add the n number and p value in the figure 13 and 15.
- The authors should add the reasons about HeLa cells and MCF-7 cells no other cellular type in the manuscript. The authors should add the normal cell type in the manuscript to demonstrate cytotoxicity could not induced by peptidomimetics in normal cells.
- The authors should add figure S59 into figure 15. And the authors also should indicate the area of dead, late apoptosis, early apoptosis, total apoptosis, and live type in the results of flowcytometry.
- The authors should add statistic results about figure 14.
- The authors should add the results of MCF-7 cells and normal cell type in the figure 14 and figure 15.
- The provide the detail mechanism of biological activity of peptidomimetics. And, the authors should provide the advantages and disadvantages of current peptidomimetics compared to past compounds from the perspective of biological mechanism.
Author Response
Respond to the Reviewer 2, Round 2.
Dear Reviewer,
Thank you for this 2nd round of review and the specific questions and comments on the biological activity of peptidomimetics. We will try to answer all the questions and accept most of your comments to improve and make the manuscript of higher quality. The answers and comments follow in this order:
- The authors should re-organize the paragraph of 3.1.5 biological activity. The section should more clearly to facilitate understanding.
Response: The minor changes to the paragraph of 3.1.5 biological activity are made to make it more understandable. The changes are highlighted in green and are therefore visible in the text.
- The authors should add the n number and p value in the figure 13 and 15.
Response: The n number and p value are added to the revised Figures 13 (p. 16) and 15 (p. 18/19), and the associated Figure captions are rewriten to reflect this update (p. 16, lines 453-455 and p. 19, lines 510-513).
In the Materials and Methods section, at the end of 3.1.5. Biological activity, we have added text with regard to statistical analysis: „The data in the graphs are expressed as mean ± standard deviation (±SD), and the error bars in the figures indicate the SD. Differences between means were analyzed using the ANOVA test, followed by post-hoc Tukey’s test. A significant difference was considered at a p value < 0.05.“ (lines 774 – 777).
- The authors should add the reasons about HeLa cells and MCF-7 cells no other cellular type in the manuscript. The authors should add the normal cell type in the manuscript to demonstrate cytotoxicity could not induced by peptidomimetics in normal cells.
Response: We have tested peptidomimetics 2-5 on HeLa and MCF-7 lines because these are human tumor cell lines that we currently have in our Lab. The normal cell type is not included in this work for two reasons: first, looking at the relevant literature, only some of the authors present the cytotoxicity of potent antitumor compounds on tumor cells vs. normal cells, most of them test compounds of interest only on human tumor cell lines. The reason for that is probably in the following fact: National Cancer Institute uses the NCI-60 Human Tumor Cell Lines Screen, which utilizes 60 different human tumor cell lines to identify and characterize new compounds with growth inhibition or killing activity toward tumor cells. Therefore, we also did not included normal cells in our research. In addition, it is not possible to perform additional experiments on normal cells in the 5 days we have to complete the revision of this manuscript. However, we will consider the reviewer's suggestion when planning the next experiments.
- The authors should add figure S59 into figure 15. And the authors also should indicate the area of dead, late apoptosis, early apoptosis, total apoptosis, and live type in the results of flowcytometry.
Response: Figure S59 has been inserted in Figure 15. The area of dead, late apoptotic, early apoptotic, total apoptotic, and live cells can be seen in this representatie histogram and is indicated by number.
- The authors should add statistic results about figure 14.
Response: There is no statistics for Figure 14, because this staning method with the fluorescent dyes FDA and PI coupled with fluorescent microscopy was applied only as a qualitative analysis. Flow cytometry was further used to quantify cell death in the population of cells.
- The authors should add the results of MCF-7 cells and normal cell type in the figure 14 and figure 15.
Response: We did not assess cell death in MCF-7 cells because HeLa cells were more sensitive toward treatment with peptidomimetics of interest in cytotoxic assays (the corresponding sentence is added at p. 15, lines 448-450). Regarding normal cell type, as explained in the response to question 3 - we did not include normal cells in our research.
- The provide the detail mechanism of biological activity of peptidomimetics. And, the authors should provide the advantages and disadvantages of current peptidomimetics compared to past compounds from the perspective of biological mechanism.
Response: First, the determination of the mechanism of action (MOA) was not the aim of the presented study, so we have not commented on it in our discussion, since it cannot be assumed solely on the basis of the results obtained here. Therefore, we believe that a discussion the advantages and disadvantages of the current peptidomimetics compared to the previous agents from the perspective of biological mechanism is beyond the results presented in this paper. In addition, the initial evaluation by Reviewer 2 stated that „the authors should constrict all paragraphs of manuscript“ and that discussion of mechanism of action, as now suggested, would only further increase the size of the entire paper.
However, we agree with your valuable comment that the mechanism of action of the peptidomimetics tested should be investigated, which will certainly be the subject of our further investigations. But, to improve the work in this sense, the discussion gives known data from the literature about the possible MOA of the peptidomimetics (p. 19, lines 525-531).

Round 3
Reviewer 2 Report
The manuscript good enough to publication